# A Lightweight Intelligent Network Intrusion Detection System Using One-Class Autoencoder and Ensemble Learning for IoT

**DOI:** 10.3390/s23084141

**Published:** 2023-04-20

**Authors:** Wenbin Yao, Longcan Hu, Yingying Hou, Xiaoyong Li

**Affiliations:** 1School of Computer Science, Beijing University of Posts and Telecommunications, Beijing 100876, China; 2Beijing Key Laboratory of Intelligent Telecommunications Software and Multimedia, Beijing University of Posts and Telecommunications, Beijing 100876, China; hulongcan@bupt.edu.cn (L.H.); houyingying@bupt.edu.cn (Y.H.); 3School of Cyberspace Security, Beijing University of Posts and Telecommunications, Beijing 100876, China; lixiaoyong@bupt.edu.cn

**Keywords:** intrusion detection, one-class classification, bidirectional GRU autoencoder, novelty detection, IoT

## Abstract

Network intrusion detection technology is key to cybersecurity regarding the Internet of Things (IoT). The traditional intrusion detection system targeting Binary or Multi-Classification can detect known attacks, but it is difficult to resist unknown attacks (such as zero-day attacks). Unknown attacks require security experts to confirm and retrain the model, but new models do not keep up to date. This paper proposes a Lightweight Intelligent NIDS using a One-Class Bidirectional GRU Autoencoder and Ensemble Learning. It can not only accurately identify normal and abnormal data, but also identify unknown attacks as the type most similar to known attacks. First, a One-Class Classification model based on a Bidirectional GRU Autoencoder is introduced. This model is trained with normal data, and has high prediction accuracy in the case of abnormal data and unknown attack data. Second, a multi-classification recognition method based on ensemble learning is proposed. It uses Soft Voting to evaluate the results of various base classifiers, and identify unknown attacks (novelty data) as the type most similar to known attacks, so that exception classification becomes more accurate. Experiments are conducted on WSN-DS, UNSW-NB15, and KDD CUP99 datasets, and the recognition rates of the proposed models in the three datasets are raised to 97.91%, 98.92%, and 98.23% respectively. The results verify the feasibility, efficiency, and portability of the algorithm proposed in the paper.

## 1. Introduction

In the digital era featured by the IoT, devices such as smart appliances, smart medical devices, and driverless cars are becoming increasingly common, thereby making work and life easier (as shown in Figure 1). However, such interconnected IoT devices have brought new cybersecurity risks during interaction, especially attacks against the IoT. Intruders enter the network system through inauthentic access, and then they modify and steal information. For example, changing the frequency of cardiac pacemakers may kill patients, remotely braking the engine and modifying instructions may cause car accidents, and writing ransomware may generate illegal profits. Attacks against the IoT will bring huge losses to users and even threaten their lives. Therefore, it is crucial to develop an efficient and safe IoT intrusion detection system for network system defense.

NIDS is a major shield for the cybersecurity of the IoT; it can audit data packets in real time and when suspicious data is found, and it serves as a network security device that gives the alarm or takes response measures. Traditional NIDS [1,2,3,4,5,6,7,8,9] aim at binary classification or multi-classification and build a model through feature engineering (PCA “Principal Component Analysis”, LDA “Linear Discriminant Analysis”, SVD “Singular Value Decomposition”, etc.) and machine learning (such as the BP neural network, CNN, RNN, SVM, etc.). Although such a model can prevent known attacks, it is not so good at guarding against unknown attacks (such as zero-day attacks). As online data grow rapidly, the ever-increasing bandwidth and traffic have put traditional intrusion detection under a lot of pressure. Massive resources needed in the in-depth detection of messages tend to overload NIDS and prolong the processing time of messages. In extreme cases, some messages might have to be discarded. If the messages with characteristics of an attack are discarded, security accidents are highly possible to occur. Therefore, this model no longer meets the real-time and accuracy requirements of current intrusion detection.

In recent years, the rapid growth of hardware such as CPU, GPU, and memory enables wider use of deep learning and ensemble learning technology in NIDS, and the recognition rate has been significantly improved. This paper proposes a lightweight intelligent network intrusion detection system based on the One-Class Bidirectional GRU Autoencoder and Ensemble Learning (OC-Bi-GRUs-AE and EL) model. The innovative highlights and major contributions of this model are as follows:The OC-Bi-GRUs-AE model proposed in this paper tackles the problem of model closure, and it is more applicable to abnormal data detection and novel data detection, thereby enabling it to effectively deal with unknown cyberattacks.The complete model with OC-Bi-GRUs-AE and EL proposed in this paper solves the imbalance of dataset types, and it can quickly recognize whether a piece of network data is an attack, as well as identify the type of attack efficiently.The method proposed in this paper is portable and shows remarkable performance in many intrusion detection datasets. In addition, the model is able to cope with unknown attacks and identify them as the type most similar to existing ones.

The remainder of this article is organized as follows. Section 2 provides a review of some related works. Section 3 elaborates on the proposed methods, Section 4 describes our experiment and result, and Section 5 is the conclusion.

## 2. Related Work

Establishing a trusted network trust system to lower network risks and guard network security is the primary function of intrusion detection systems. Many classification algorithms based on supervised learning, such as the deep neural network [10] and SVM [11], have been widely used in IDS tasks in the past, and are able to perform binary and multi-classification tasks satisfactorily. These algorithms are good at detecting known attack types. For instance, Zhang [10] employed a denoising autoencoder with a weighted loss function for feature selection and then classified the selected data by MLP for intrusion identification. With a small feature selection ratio of 5.9%, the proposed scheme delivered great performance according to different criteria. Safaldin [11] proposed the modified binary grey wolf optimizer with SVM, therein aiming to increase intrusion detection accuracy, detection rate, and to reduce processing time in the WSN environment. The results showed that the proposed method with seven wolves greatly outperformed other comparative algorithms.

One-Class Classification is an anomaly detection algorithm. In the dataset, if the volume of one type of data is too large, this certain type will be used for training to split from other data. In an intrusion detection dataset, there are usually many pieces of normal network data, but little attack data. It is efficient and fast to detect data anomalies with One-Class Classification. The commonly used algorithms include Meta-Learning [12], the Interpolated Gaussian Descriptor [13], the OCSVM [14,15,16,17,18,19], and the Autoencoder [20,21,22,23,24,25,26]. Among the above-mentioned algorithms, the OCSVM shows a high detection rate in small sample datasets, especially when Ghada [19] improved the performance of OCSVM anomaly-based machine-learning-enabled intrusion detection systems by tuning hyperparameter optimization techniques. In this case, an efficient, scalable and distributed intelligent IDS was built to detect intrusion in the IoT, and the model was evaluated by Ensemble Learning optimization technology. A comparative analysis was performed on the performance and predictability of intrusion detection models in the IoT. As the number of datasets grows and the functions of hardware and software improve, Automatic Encoder algorithms are being more widely adopted. Song [24] designed a stacked self-encoder model with a focus on the model capacity, depth, and the size of the middle layer that represents the compressed latent information of the given data. The results of the experiment showed that, the larger the model size is, the better and more stable the performance of the stacked self-encoder model will be, and the selection of the latent size can improve performance as well.

Ensemble Learning [27,28,29,30,31,32,33] solves problems by training multiple learners and combining them. Ensemble is better at generalization than weak learners, and can turn weak learners that are only slightly better than random guess into strong learners with accurate prediction. Khan [30] proposed a novel intrusion detection approach for the IoT based on an ensemble voting classifier that combines multiple traditional classifiers as a base learner. Saba [32] proposed a two-stage hybrid method, selected appropriate features using the genetic algorithm, employed an ensemble classifier, and applied SVM and decision tree to mark the attack as malicious or normal. Yao [33] proposed a two-layer soft-voting ensemble learning model with RF, lightGBM and XGBoost as base classifiers, and used the adversarial validate algorithm to test the consistency of the data distribution in training and testing dataset to determine whether the dataset needs re-splitting. The results showed that the model has a higher accuracy rate in both binary and multi-classification than other One-Class Classification models.

To make intrusion detection systems smarter and more accurate, we propose a lightweight intelligent network intrusion detection system using a One-Class Autoencoder and Ensemble Learning for the IoT (our research domain is shown in Figure 2). The proposed approach focuses on defending against unknown zero-day attacks and identifying such attacks as the type most similar to existing known attacks, thereby solving the problem of a low recognition rate caused by the imbalance of data types in traditional datasets. Our goal is that the proposed NIDS has the advantages of high performance, high prediction accuracy, and portability. A comparision of our approach and related work is shown in Table 1.

## 3. Proposed Methods

### 3.1. One-Class Bidirectional GRU Autoencoder

In this paper, we propose a One-Class model based on a Bi-GRUs-AE for anomaly detection (as shown in Figure 3). The model builds a framework with an Autoencoder, and its structure includes two parts—an Encoder and a Decoder. The Encoder realizes dimensionality reduction by transforming data from a high-dimensional space into a low-dimensional space. The Decoder achieves dimensionality increases by transforming data from a low-dimensional space into a high-dimensional space. With the Bi-GRUs network and optimization method, the Autoencoder plays a supervisory role through inputting data to guide the Bi-GRUs network in trying to learn about the map. In this way, a reconstructed output is achieved, which ensures that the output data share the same dimension and similar content as the input data.

In the case of One-Class classification, the model makes full use of the data correlation of the Autoencoder. Since the Autoencoder model trained by normal data is related to normal data, there will be a big loss between the output and input data in the case of abnormal data. The range of the loss is used for data classification to determine whether the input data fall into a certain type. During training, the following Equations (Equation 1) and (Equation 2) are used to compare the input data with the output data:(1)$$loss_{mae}=\sum_{i=1}^{N-1}\left|x_{i}-x_{i}^{\prime }\right|$$
(2)$$loss_{mse}=\sum_{i=1}^{N-1}{\left(x_{i}-x_{i}^{\prime }\right)}^{2}$$

In the above equations, $x_{i}$ represents the i-th data in the input data sequence, and $x_{i}^{\prime }$ represents the i-th data in the output data sequence after Autoencoder training. The value “mae” means “Mean Absolate Error”, and “mse” means “Mean Square Error”.

Within the Autoencoder, the Gate Recurrent Unit (GRU) is the smallest network structure unit, as shown in Figure 4. It contains two channels of input and one channel of output, and the internal output is mainly obtained by controlling the calculation of the reset gate and update gate.
(3)$$r_{t}=sigmoid\left(W_{r}\cdot \left[h_{t-1},x_{t}\right]\right)$$
(4)$$z_{t}=sigmoid\left(W_{z}\cdot \left[h_{t-1},x_{t}\right]\right)$$
(5)$$\tilde{h_{t}}=tanh\left(W*\left[r_{t}⊗h_{t-1},x_{t}\right]\right)$$
(6)$$h_{t}=\left(1-z_{t}\right)⊗h_{t-1}+z_{t}⊗\tilde{h_{t}}$$

Among them, $x_{t}$ represents the input information at time t, $h_{t-1}$ means the output information at time t−1; sigmoid and tanh are commonly used activation functions in neural networks; $r_{t}$ means the reset gate, $z_{t}$ means the update gate, $\tilde{h_{t}}$ means the candidate hidden state, ⊗ means the Hadamard matrix, and $h_{t}$ means the output data. $r_{t}$ decides how to combine the new input information with the previous memory $h_{\left(t-1\right)}$. The larger the value of $r_{t}$ is, the larger the memory needed for the last moment. When the value of $r_{t}$ approaches 1, it means the hidden state of the last moment is kept. When its value approaches 0, it means that all the contents of the last moment need to be discarded. $z_{t}$ controls the extent to which the state information of the last moment is brought into the current state, that is, the update gate helps the model in deciding how much information from the past is to be transmitted to the future. The closer it is to 1, the more data are “memorized”; the closer it is to 0, the more data are “forgotten”.

Given that GRU constitutes a type of Forward memory, such memory can only be obtained from a past moment, but not the future. Therefore, in this paper, two GRUs are combined to form Bi-GRUs, in which one GRU adopts Forward memory and the other adopts Backward memory, and both of them are connected to an output layer. This structure provides each point in the input sequence of the output layer with complete context information of the past and future, as shown in Figure 5.

To ensure that the OC Bi-GRUs-AE model can recognize unknown abnormal data, in the process of splitting the dataset, the training set is all normal data, while the evaluating set contains normal and abnormal data in 1:1 proportion. After the Bi-GRUs-AE is trained with the training set to generate a stable model, the difference range between the data generated by the model and the input data is obtained. In the testing process, the evaluating set and the data generated by the model are used for discrimination. If the difference is within the range of normal data, the data will be identified as normal, or abnormal. The algorithm is as shown in Algorithm 1.
**Algorithm 1:** One-Class LossInput: Training set, # Normal data            Evaluating set, # Normal data and Unnormal data = 1:1Output: Normal label, Unnormal labelProcess:#step 1: Bi-GRU AE train         model = Bi-GRU AE) # init model         model.fit(TrainData, split = 0.2, batchsize, epoch)        PredictData = model.predict(TrainData)        Loss = abs(TrainData-PredictData)#absolute value        Loss = sort(Loss)        Loss_train = max(Loss)        Return Loss_train# step 2: One-Class Classification        PredictData = model.preict(TestData)        Loss = abs(TrainData-PredictData)        Loss = sort(Loss))        Loss = max(Loss)        If Loss > Loss_train:           Output: Unnormal label        Else:           Output: Normal label

### 3.2. Ensemble Learning

This paper proposes a Soft-Voting Ensemble model to improve the recognition rate of multi-classification data, in which Random Forest, XGBoost, and LightGBM are used as base classifiers, and Soft-Voting technology is used to vote on the prediction results of the three classifiers for optimal classification. Especially in the case of unknown attack types, they can be identified as the type most similar to existing known attacks through voting. Soft Voting [30,34] is an algorithm used to calculate and vote on the probability output generated by the base classifiers. Simple Soft Voting treats the probability output of each base classifier equally, as demonstrated in Equation (Equation 7), while weighted Soft Voting weights the base classifiers or the types, as demonstrated in Equation (Equation 8). In the process of Soft Voting, the individual classifier $p_{i}$ outputs a K-dimensional vector ${\left(p_{i}^{1}\left(x\right)...p_{i}^{k}\left(x\right)\right)}^{T}$ to the data X, $p_{i}^{k}\left(x\right)\in \left[0,1\right]$, where $w_{i}$ means the weight of the classifier $p_{i}$.
(7)$$p^{j}\left(x\right)=\frac{1}{T}\sum_{i=1}^{T}p_{i}^{j}\left(x\right)$$
(8)$$p^{j}\left(x\right)=\sum_{i=1}^{T}w_{i}^{}p_{i}^{j}\left(x\right)$$

In terms of the selection of base classifiers, Random Forest [35] introduces random feature selection based on a decision tree, and its performance on generalization can be further improved by the increased difference among individual learners. XGBoost (Extreme Gradient Boosting) [36] adopts the level-wise strategy to grow the decision tree, and applies a second-order Taylor polynomial to the loss function based on GBDT. In each iteration, a strategy similar to Random Forest is adopted, which allows data sampling and can significantly improve the speed and efficiency. LightGBM (Light Gradient Boosting Machine) [37] adopts the leaf-wise strategy to grow the decision tree, which allows efficient parallel training, faster training speed, lower memory consumption, and higher accuracy.

As shown in Figure 6, a lightweight intelligence NIDS includes data processing, One-Class Classification, and multi-classification. In data processing, first the original dataset is standardized, so that all the data are within the range of 0 to 1. Second, since the dataset contains some redundant features, which do not work well in model identification, we adopt the feature extraction method. The dataset after feature extraction is better positioned for model identification. Finally, we divide the dataset into the training set and evaluation set. The training set contains only normal data, while the evaluation set contains both normal and attack data in a 1:1 proportion. In the One-Class Classification, we use the training set for the OC-Bi-GRUs Autoencoder model. After it is stabilized, the model returns a loss in a very small range; in the case of the anomaly data, a loss in another range is returned. Besides accurately identifying normal and abnormal data, the OC-Bi-GRUs Autoencoder model can identify unknown attacks as abnormal. This solves the imbalance in data types for traditional binary classification or multi-classification. In terms of multi-classification, we apply Soft-Voting Ensemble Learning. By comprehensively analyzing the results of various learners, we manage to improve the classification accuracy of abnormal attacks and are able to identify them as the type most similar to known attacks.The proposed models focus on defending against unknown zero-day attacks and identify such attacks as the type most similar to existing known attacks, thereby solving the problem of low recognition rate caused by the imbalance of data types in traditional dataset. The proposed NIDS has the advantages of high performance, high prediction accuracy, and portability.

## 4. Experiment and Result

### 4.1. Dataset

The WSN-DS dataset [38] is an intrusion detection dataset specially designed for wireless sensor networks. It contains 374,661 pieces of data, and each piece consists of 19 types of feature data (18 feature labels and a One-Classification label). The classification label contains one type of normal data and four types of DoS attacks: Grayhole, Blackhole, TDMA, and Flooding.

The UNSW-NB15 dataset [39,40] is a comprehensive dataset for NIDS, which was created at the Cyber Range Lab of the Australian Center of Cyber Security in 2015. It features a hybrid of the real modern normal and the contemporary synthesized attack activities of the network traffic by utilizing the IXIA PerfectStorm tool. This dataset contains 2,540,044 pieces of data, which include normal data and nine types of attack: Exploits, Fuzzers, Reconnaissance, Generic, DoS, Analysis, Backdoors, Shellcode, and Worms.

The KDD CUP99 dataset is a network dataset created by the Defense Advanced Research Projects Agency (DARPA) in the MIT Lincoln Laboratory in 1998, which was a simulation of the local area network (LAN) of the United States Air Force. After some processing, KDD CUP99 entered the Third International Knowledge Discovery and Data Mining Tools Competition (KDD Cup). This dataset contains 4,898,430 pieces of data. Each piece contains 41 feature labels and One-Classification label, and the attack falls into one of the given types: U2R, DoS, R2L, or probing.

### 4.2. Feature Extraction, Dataset Split, and Metrics

In terms of the features of the original network intrusion detection dataset, there are problems such as inconsistent standards, duplicate data, and null value data. Therefore, such a dataset cannot be used directly for model training; instead, they need to be processed by feature engineering. First, assign a value to the null value data in the dataset and replace it with 0. Second, replace label features in the dataset with numbers. In the KDD99 dataset, protocol_type contains three kinds of label data: TCP, UDP, and ICMP; service contains 70 types of label data, such as HTTP, FTP, and SMTP; and flag contains 11 types of label data, such as REJ and RSTR. In the UNSW-NB15 dataset, proto contains 133 types of labels such as TCP, UDP, and IPv6; service contains 13 types of labels, such as HTTP, FTP, and DNS; and state contains 11 types of labels such as FIN, RST, and ACC. The above-mentioned types of data were each replaced with numbers. Last, all features in the dataset were standardized. The Min–Max normalized method was used to standardize the data, which was scaled between 0 and 1.
(9)$$x_{new}=\frac{x-x_{min}}{x_{max}-x_{min}}$$

There tend to be redundant or irrelevant features in the feature set. Feature analysis enables important features in the feature set to be extracted for higher speed and accuracy of model training. Nour Moustafa [41] compared the efficiency and reliability of the UNSW-NB15 and KDD99 in terms of features to distinguish between normal and abnormal records, introduced an association rule mining algorithm in feature selection to generate the strongest features, and reduced the computational time from the KDD99 and UNSW-NB15 dataset. The experimental results showed that the evaluation criteria of the replicated UNSW-NB15 features of the KDD99 dataset were better than the original KDD99 features. Janarthanan, T. and Zargari, S. [42] employed data mining and machine learning techniques to explore significant features in detecting network intrusions, proposed a subset of features, reduced resource consumption, and maintained high detection rates. Dong R H [43] applied the information gain ratio method in selecting the feature of WSN-DS, which reduced the computational complexity of the intrusion detection method and cut the computation and time overhead in detection.

In this paper, the Gini index Equation (Equation 10) of Random Forest was used to determine the importance of features. Firstly, the contribution of each feature in each decision tree was calculated, and the difference of the Gini Index before and after the branch of the feature at a certain node was obtained. Then, the contribution of each feature was normalized and sorted according to their contribution.
(10)$$GI_{q}^{\left(i\right)}=\sum_{c=1}^{N}p_{qc}^{\left(i\right)}*\left(1-p_{qc}^{\left(i\right)}\right)=1-\sum_{c=1}^{N}{\left(1-p_{qc}^{\left(i\right)}\right)}^{2}$$

In Equation (Equation 10), *c* represents the number of types, and $p_{qc}$ means the proportion of type *c* in node *q*. The VIM (variable importance measures) of the feature $x_{j}$ in node *q* of the *i*-th tree are as follows:(11)$$VIM_{jq}^{\left(GI)(i\right)}=GI_{q}^{\left(i\right)}-GI_{l}^{\left(i\right)}-GI_{r}^{\left(i\right)}$$

If the node of the feature $x_{j}$ in the decision tree is in set Q, the VIM of $x_{j}$ in the i-th tree are as follows:(12)$$VIM_{j}^{\left(GI)(i\right)}=\sum_{q\in Q}VIM_{jq}^{\left(GI)(i\right)}$$

If the number of trees in the Random Forest is M, then:(13)$$VIM_{j}^{\left(GI\right)}=\sum_{i=1}^{M}VIM_{jq}^{\left(GI)(i\right)}$$

Finally, all the VIM are normalized:(14)$$VIM_{j}^{\left(GI)(i\right)}=\frac{VIM_{j}^{\left(GI\right)}}{\sum_{i=1}^{N}VIM_{i}^{\left(GI\right)}}$$

The VIM of features of Random Forest are used to extract features from KDD99, UNSW-NB15, and WSN-DS datasets, respectively. Figure 7, Figure 8 and Figure 9 rank the VIM of the features of the three datasets. According to the importance of features, we tried to extract the most important N features to reduce invalid ones and improve the recognition rate of the final model. Table 2 contains the features extracted from the WSN-DS, KDD99, and UNSW-NB15 datasets. With multiple tests, we extracted 20 features from the KDD99 and UNSW-NB15 and 14 features from the WSN-DS for subsequent model training.

In terms of the dataset split, the training set contained only normal data, while the evaluating set contained both normal and attack data in a 1:1 proportion. Table 3 shows the data split results of the three datasets.

The confusion matrix (Table 4) is a visual tool in supervised learning, which is mainly used for comparing the classification results and the real information of instances. Each row in the matrix represents a real category, and each column represents a predicted category of the instance.

True Positive (TP): Attack data predicted as an attack.

False Positive (FP): Normal data predicted as an attack.

True Negative (TN): Normal data predicted as normal.

False Negative (FN): Attack data predicted as normal.

Based on the confusion matrix, we can evaluate the performance of our proposed model.

Accuracy: For the proportion of correctly predicted samples to the total samples, the value range is [0, 1]. The higher the value is, the better the model will perform in its prediction in terms of accuracy. The calculation is as follows:(15)$$Accuracy=\frac{\left|TP\right|+\left|TN\right|}{\left|TP\right|+\left|TN\right|+\left|FP\right|+\left|FN\right|}$$

Precision: For the proportion of correct predictions to all predicted “attack” samples, the value range is [0, 1]. The higher the value is, the better the model will perform in its prediction in terms of precision. The calculation is as follows:(16)$$Precision=\frac{\left|TP\right|}{\left|TP\right|+\left|FP\right|}$$

Recall: The proportion range of correctly predicted “attack” samples in true attack labels is [0, 1]. The higher the value is, the better the model will perform in its prediction in terms of recall. The calculation is as follows:(17)$$Recall=\frac{\left|TP\right|}{\left|TP\right|+\left|FN\right|}$$

F1_score: a weighted harmonic mean of the model precision and recall, which serves as a derived measurement for effectiveness. The calculation is as follows:(18)$$F1\_score=2\times \frac{Precision\times Recall}{Presion+Recall}$$

In this chapter, we processed the dataset by feature engineering. First of all, the Min–Max normalized method was used to standardize the original dataset so that all the data were within the range of 0 to 1. Second, as the dataset of KDD99, UNSW-NB15 and WSN-DS contains some redundant features, the Gini index of Random Forest was used to extract features. With multiple tests, we extracted 20 features from the KDD99 and UNSW-NB15 and 14 features from the WSN-DS for subsequent model training. Third, we divided the dataset into the training set and evaluation set. The training set contained only normal data, while the evaluating set contained both normal and attack data in a 1:1 proportion. Finally, we use some well-established standard metrics (such as Accuracy, Precision, Recall, and F1_score) to evaluate the performance of our approach.

### 4.3. One-Class Classification

We ran our model on a workstation with Intel(R) Xeon(R) Silver 4210R CPU, NVIDIA GeForce RTX 3090 GPU 24 GB, 50 GB RAM, 50 GB HD, Ubuntu 18.04.5 OS, and all the tasks were performed using Python 3.7 with scikit-learning (version = 0.24.2).

The model of One-Class Autoencoder in this paper is shown in Figure 10 and Figure 11, which adopts seven Bi-GRUs, three Denses and two Dropouts. There were 20 features in the input layers of the KDD99 and UNSW-NB15 dataset, and 14 in the input layer of the WSN-DS dataset. Both the KDD99 and UNSW-NB15 contained 20 features after feature extraction. These 20 features were encoded and decoded by the Autoencoder. The encoder compressed the 20 features into 8 through three Bi-GRUs: (20,1) → (20,16) → (20,8) → (8), and then deleted some information by Dropout to prevent overfitting. The Decoder converted 8 features into 20 through four Bi-GRUs, (8,1) → (8,8) → (8,16) → (8,20) → (20), then deleted some information by Dropout to prevent overfitting, and, finally, 20 features were obtained with sigmoid. For the WSN-DS dataset, we adopted the same network structure, and only the input and output were adjusted. The Encoder compressed 14 features into 4 through three Bi-GRUs: (14,1) → (14,7) → (14,4) → (4), then deleted some information by Dropout to prevent overfitting, and, finally, 4 features were obtained with sigmoid. The Decoder converted 4 features into 14 features, (4,1) → (4,4) → (4,7) → (4,14) → (14), then deleted some information by Dropout to prevent overfitting, and, finally, 20 features were obtained with sigmoid.

Our model introduced “adam” as the optimizer and “mae” as the loss function. Based on the number of training sets and the system configuration, during the training of the KDD99 and UNSW-NB15 training sets, the batch_size was set to 10,000, while that for the WSN-DS training set was 2048. A higher value of batch_size can avoid overfitting of the model. Table 5 shows some related information of the three datasets after training.

We used the trained model to obtain the threshold value of the training set, obtain the difference between the generated data and the training data, and take the absolute value. Then, we calculated the maximum value, a, for each item of different data, and this value generated from the normal data remained in a certain range (as shown in Figure 12, Figure 13 and Figure 14). The difference thresholds of the three datasets were acquired, which were then tested on the evaluating set. We obtained the difference between the generated data and the test data and took the absolute value. We then obtained the maximum value, b, for each item of different data. If b was larger than a, it was considered as a piece of abnormal data. As shown in Figure 15, Figure 16 and Figure 17, the green line indicates normal data, and the red one indicates abnormal data.

In the stage of data prediction, a reasonable threshold was selected for discrimination. All values below the threshold were set to 0, and those above the threshold were set to 1. In this way, the One-Class algorithm could be turned into a binary classification algorithm and then used for evaluating the model. Table 6, Table 7 and Table 8 are the confusion matrixes of the evaluating set.

Table 9 shows the accuracy of three datasets. With this model, the accuracy of WSN was 97.91%,that of UNSW-NB15 was 98.92%,and that of the KDD99 reached 98.23%. Experiments show that the One-Class Bi-GRUs AE algorithm presents an efficient way to identify intrusion detection data, and it can also identify abnormal data of zero-day attacks. Table 10 shows the comparison of several NIDS approaches.

### 4.4. Zero-Day Attacks Detection

The multi-class classification model of intrusion detection aims to detect the type of attack to provide tailor-made solutions for abnormal attacks. This paper proposes a new method to identify unknown abnormal attacks (zero-day attacks), which divides an abnormal dataset into a training set, an evaluating set, and a novelty set (zero-day attacks set). The type of data in the novelty set were not included in the training set and evaluating set. The training set is for generating a stable ensemble learning model; the evaluating set is for evaluating the accuracy of the model in identifying existing attack types; and the novelty set is for categorizing the attack as the type most similar to known attacks. Although the attack can only be identified as a known type, this provides a reference for the unknown types.

In the WSN-DS attack dataset, we took the attack data of the Flooding type as a novelty set (zero-day attacks set), and randomly divided the attack data of Gravhole, Blackhole, and TDMA into 80% of the training set and 20% of the evaluating set, as shown in Table 11. In the KDD99 attack dataset, we took the attack data of the Privilege type as a novelty set, and randomly divided the attack data of DoS, Probe, and Access into 80% of the training set and 20% of the evaluating set. In the UNSW-NB15 dataset, we took the attack data of the Worms type as a novelty set, and randomly divided the other eight types of attack data into 80% of the training set and 20% of the evaluating set.

During the training of the model, three learning algorithms, namely lightGBM, XGBoost and Random Forest, are adopted as base classifiers, and the Soft-Voting Ensemble Learning algorithm was used for the ensemble of the three base classifiers. For the WSN-DS dataset, the size of the Soft-Voting Ensemble Learning model stood at 52.1 MB, and the parameters of the lightGBM were as follows: 1000 for n_estimators, 0.02 for learning_rate, 0.8 for subsample, and 10.3 MB for the final model size; the parameters of XGBoost were 500 for n_estimators, 0.03 for learning_rate, 0.1 for gamma, 0.8 for subsample, and 9.2 MB for the final model size; the parameters of RandomForest were 50 for n_estimators, True for the oob_score, and 6.6 MB for the final model size. For the UNSW-NB15 dataset, the size of the ensembled model was 76 MB. The parameters of the lightGBM were 300 for n_estimators, 0.1 for learning_rate, and 8.5 MB for the final model size; the parameters of XGBoost were 100 for n_estimators, 0.1 for learning_rate, 0.1 for gamma, 0.8 for subsample, and 12.8 MB for the final model size; the parameters of RandomForest were 10 for n_estimators, True for the oob_score, and 16.7 MB for the final model size. For the KDD99 dataset, the size of Soft-Voting Ensemble Learning model tood at 21.5 MB, the parameters of the lightGBM were 300 for n_estimators, 0.02 for learning_rate, 0.8 for subsample, and 3.2 MB for the final model size; the parameters of XGBoost were 100 for n_estimators, 0.03 for learning_rate, 0.1 for gamma, 0.8 for subsample, and 1.5 MB for the final model size; the parameters of RandomForest were 50 for n_estimators, True for the oob_score, and 6.1 MB for the model size. As a result, the classification accuracy of the model from the three dataset in the training set and the evaluating set was between the maximum and minimum of the accuracy of the three types of weak classification (as shown in Table 12).

The trained model was applied to novelty detection (as shown in Table 13 and Table 14). For the novel data of the “Flooding” type assumed in the WSN-DS dataset, they could be predicted as various types of attacks (mainly “Dos” and “Access”) by three basic classifiers, while the prediction of the Soft-Voting model was more convincing. For the novelty data of the “WORMS” type assumed in the UNSW-NB15 dataset, among the three basic classifiers, most of the attack types were predicted as “Exploits”, “Fuzzers”, and “Generic”. This means that the “WORMS” type data structure is similar to the above three structures. For the novel data of the “Privilege” type assumed in the KDD99 dataset, they were predicted as attack data of the “Access” type by all three base classifiers and the Ensemble model, which indicated that this type of data was closer to the attack data of the “Access” type. By the ensemble of the three base classifiers, the final model showed better performance in its prediction.

## 5. Conclusions

In this paper, the Bidirectional GRU Autoencoder and Ensemble Learning method was adopted for novelty detection in network intrusion detection systems. Traditional intrusion detection based on binary classification was replaced by One-Class detection through modeling known normal data, which thus avoids the imbalance of dataset type caused by the small amount of abnormal data. In the One-Class Bidirectional GRU Autoencoder Model, the data correlation principle of automatic encoders was applied, where only those data similar to the training data were compressed. After the judgment was made by the model, the normal data returned a loss in a very small range, while the abnormal data returned a big loss. According to the value of the loss value, whether the network data were attack data or not could be identified. To accurately determine the anomaly type of the detected abnormal data, this paper adopted the Ensemble Learning model for Soft-Voting classification of the anomaly type identified by various base classifiers so that the unknown anomaly types (such as zero-day attacks) could be recognized as a known anomaly type as accurate as possible. The model adopted in the experiment is portable, and it delivered remarkable performance on the WSN-DS, UNSW-NB15, and KDD99 datasets. Meanwhile, this model can deal with unknown attacks and provide better approximation and accuracy for real unknown hypotheses. The recognition rates of the models for the WSN-DS, UNSW-NB15, and KDD99 datasets were 97.91%, 98.92%, and 98.23% respectively, which are much higher compared with traditional intrusion detection methods. For future work, we will consider using this model to further improve the accuracy and evaluate its performance in a distributed computing environment (such as Ray).

## Figures and Tables

**Figure 1 sensors-23-04141-f001:**
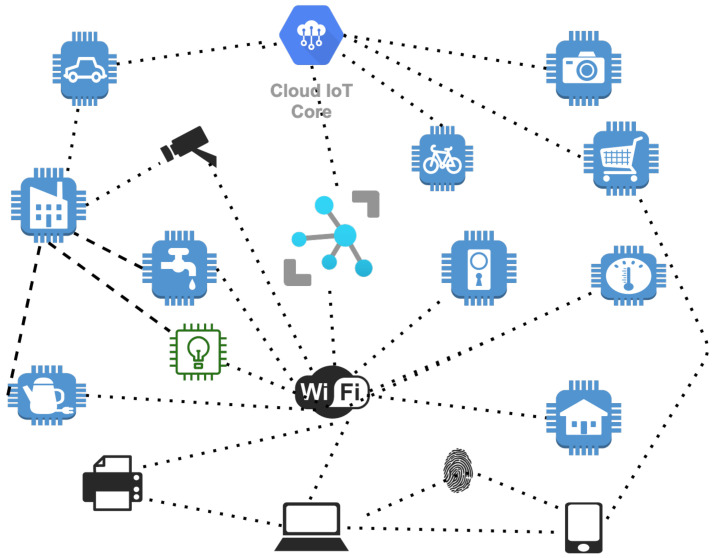
Intelligent internet life.

**Figure 2 sensors-23-04141-f002:**
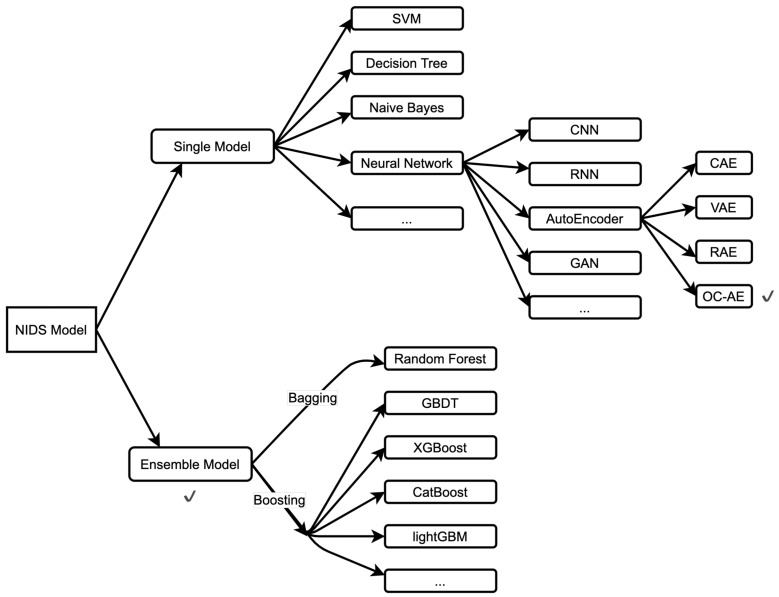
NIDS model classification (our research domain is the OC-AE. Furthermore, we use the Ensemble model).

**Figure 3 sensors-23-04141-f003:**
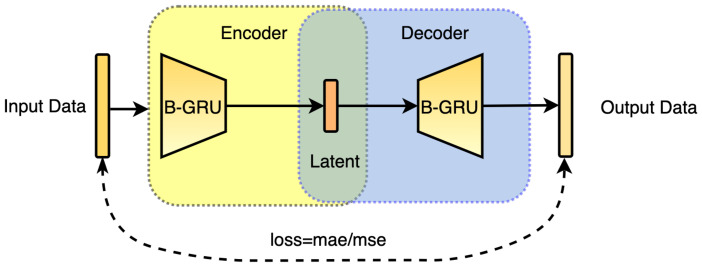
Bi-GRUs Autoencoder.

**Figure 4 sensors-23-04141-f004:**
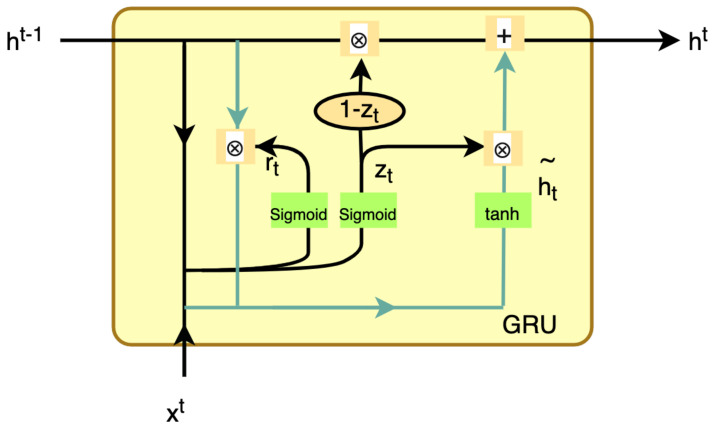
Gate Recurrent Unit (GRU).

**Figure 5 sensors-23-04141-f005:**
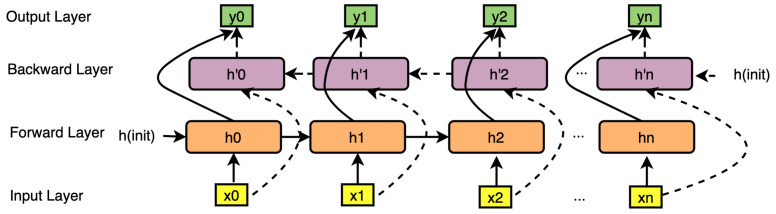
Bidirectional Gate Recurrent Units (Bi-GRUs).

**Figure 6 sensors-23-04141-f006:**
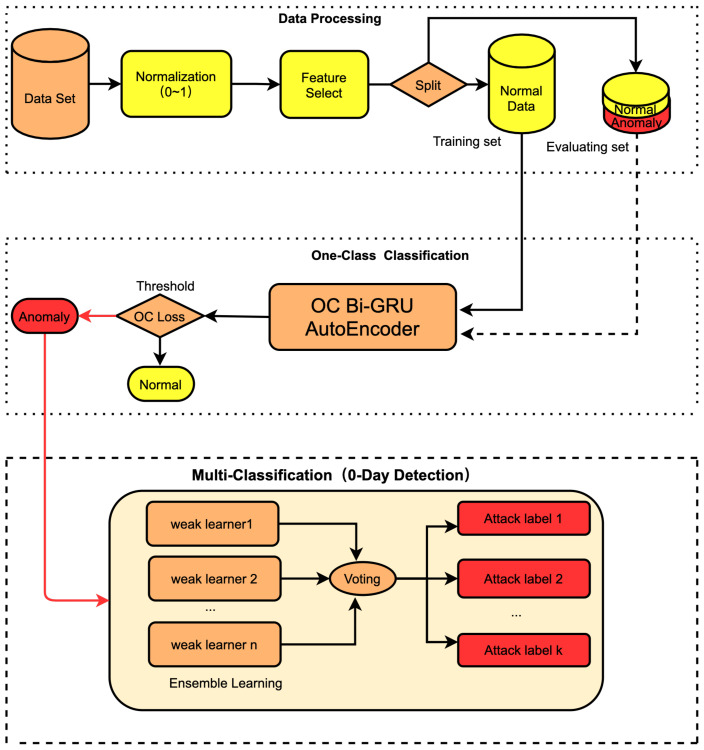
Lightweight intelligent NIDS.

**Figure 7 sensors-23-04141-f007:**
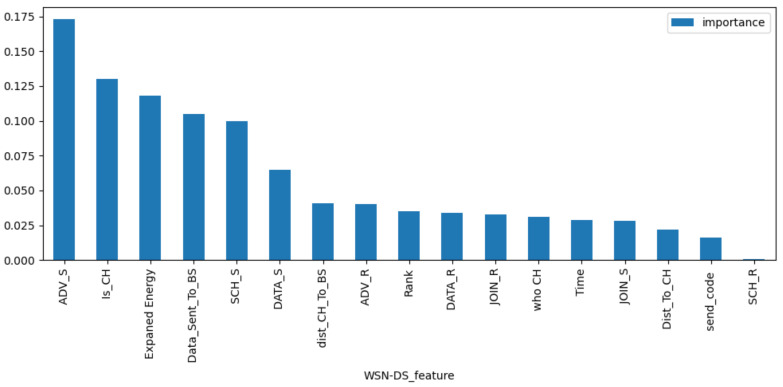
Feature importance of WSN-DS.

**Figure 8 sensors-23-04141-f008:**
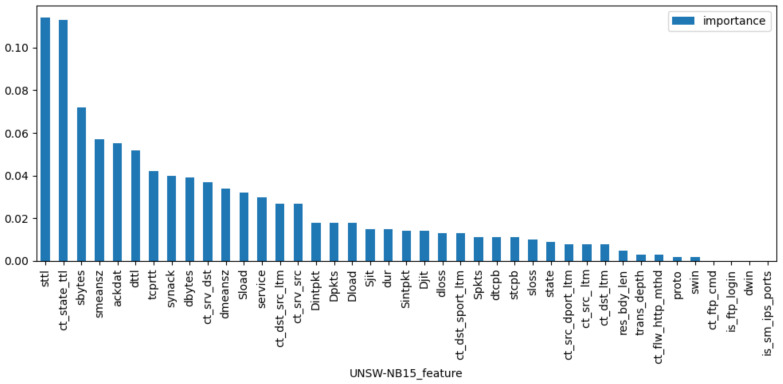
Feature importance of UNSW-NB15.

**Figure 9 sensors-23-04141-f009:**
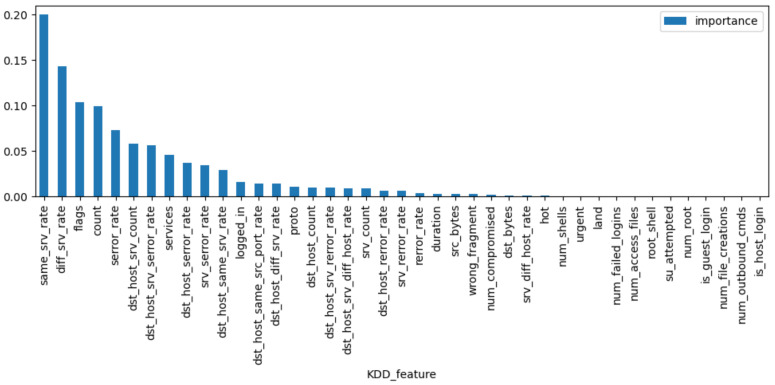
Feature importance of KDD99.

**Figure 10 sensors-23-04141-f010:**
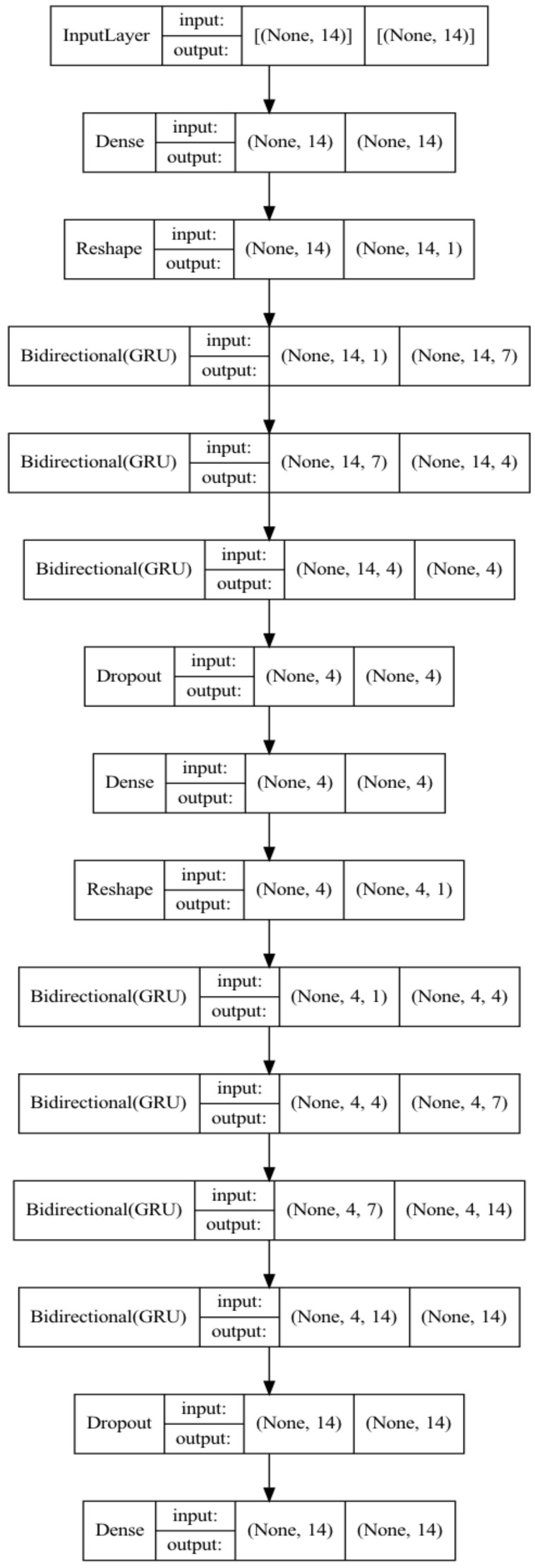
Autoencoder model of WSN-DS.

**Figure 11 sensors-23-04141-f011:**
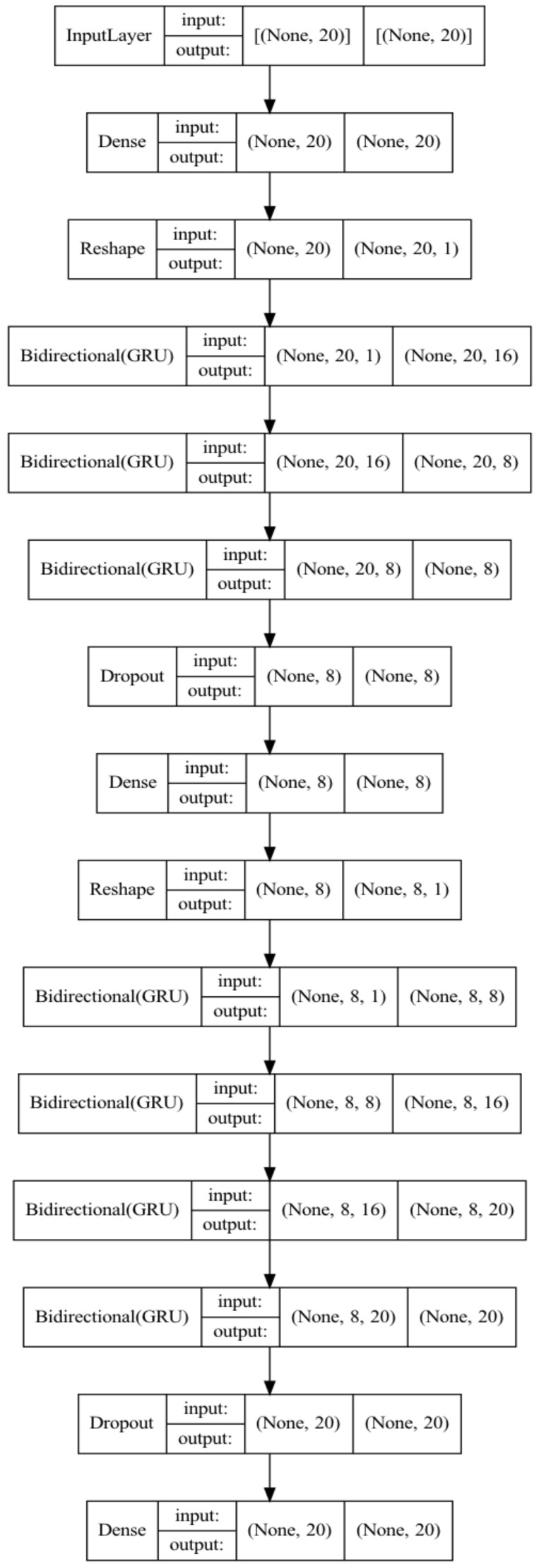
Autoencoder model of UNSW-NB15 and KDD99.

**Figure 12 sensors-23-04141-f012:**
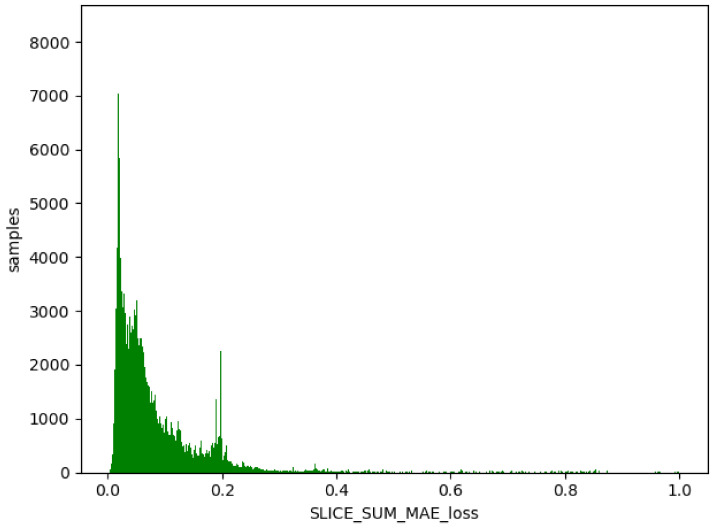
WSN-DS training set. Green indicates normal data.

**Figure 13 sensors-23-04141-f013:**
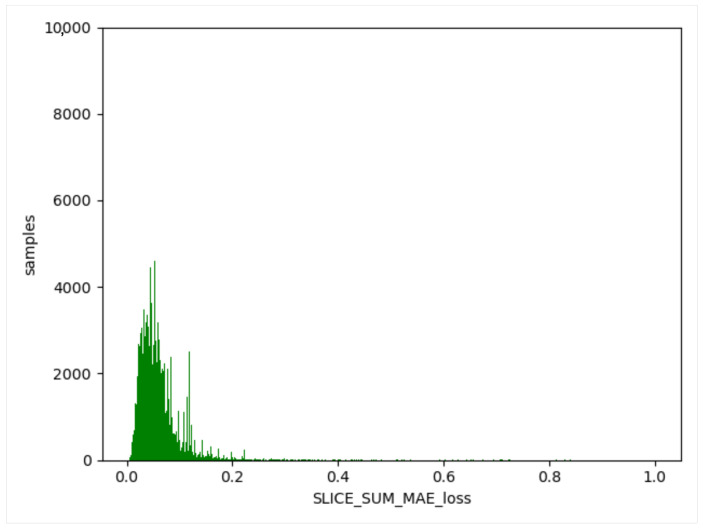
UNSW-NB15 training set. Green indicates normal data.

**Figure 14 sensors-23-04141-f014:**
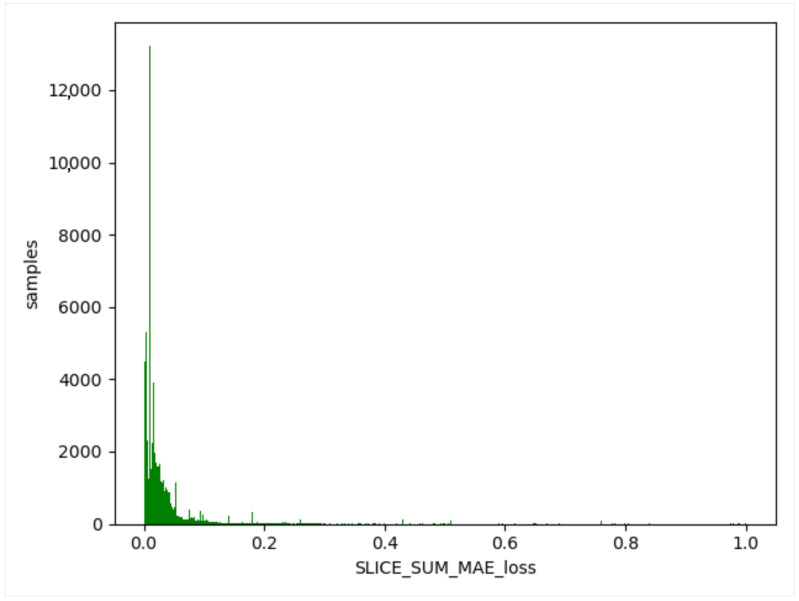
KDD99 training set. Green indicates normal data.

**Figure 15 sensors-23-04141-f015:**
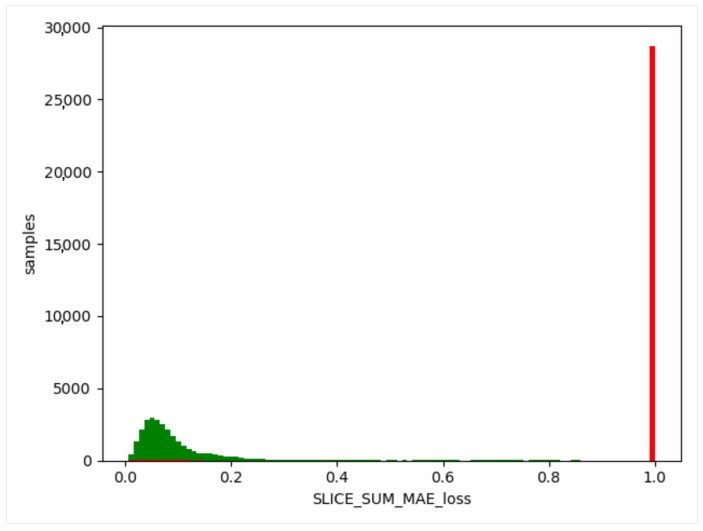
WSN-DS evaluating set. Green indicates normal data and red indicates abnormal data.

**Figure 16 sensors-23-04141-f016:**
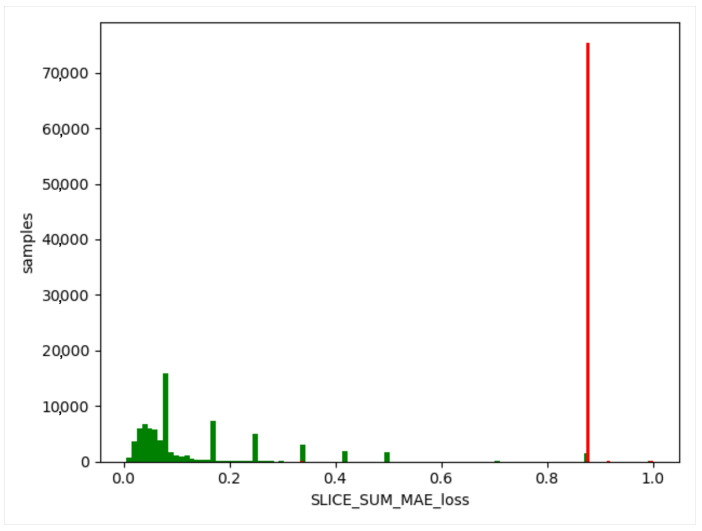
UNSW-NB15 evaluating set. Green indicates normal data and red indicates abnormal data.

**Figure 17 sensors-23-04141-f017:**
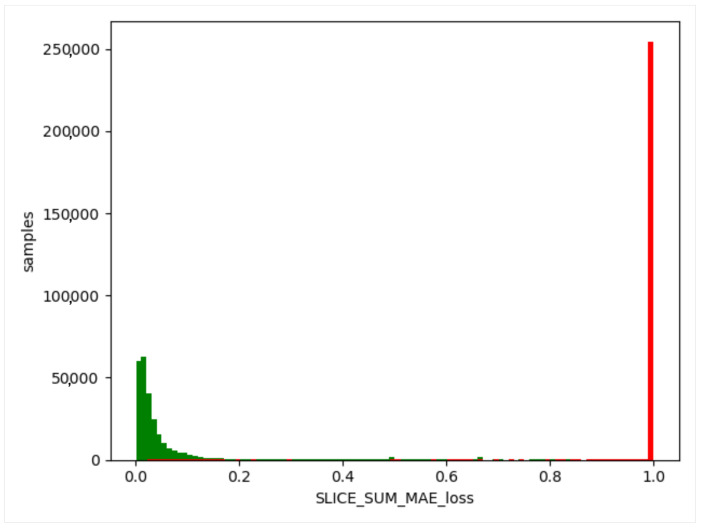
KDD99 evaluating set. Green indicates normal data and red indicates abnormal data.

**Table 1 sensors-23-04141-t001:** Comparision of our approach and related work.

Study	Method	Feature Selection	Balanced Data	Novelty (Zero-Day) Detection
Azizjon, M. [2]; Mahalakshmi, G. [3]	CNN	No	No	No
Yu, Y. [4]	FSL	No	Yes	No
Sms, A. [5]	RNN(LSTM)	No	Yes	No
Yuan, D. [6]	GAN	No	Yes	No
Safaldin, M. [11]	SVM	Yes	No	No
Abdelmoumin, G. [19]	OCSVM	Yes	No	No
Song, Y. [24]	Stacked self-encoder	No	No	No
Khan [30]	Voting Ensemble	No	No	No
Yao, W. [33]	Soft-Voting Ensemble	No	Yes	No
Our approach	OCAE + Ensemble	Yes	Yes	Yes

**Table 2 sensors-23-04141-t002:** Feature extraction.

ID	WSN-DS	UNSW-NB15	KDD99
1	Time	dur	proto
2	Is_CH	sbytes	services
3	who CH	dbytes	flags
4	Dist_To_CH	sttl	src_bytes
5	ADV_S	dttl	logged_in
6	ADV_R	Sload	count
7	JOIN_R	Dload	srv_count
8	SCH_S	smeansz	serror_rate
9	Rank	dmeansz	srv_serror_rate
10	DATA_S	Sjit	same_srv_rate
11	DATA_R	Sintpkt	diff_srv_rate
12	Data_Sent_To_BS	Dintpkt	dst_host_count
13	dist_CH_To_BS	tcprtt	dst_host_srv_count
14	Expaned Energy	synack	dst_host_same_srv_rate
15		ackdat	dst_host_diff_srv_rate
16		ct_state_ttl	dst_host_same_src_port_rate
17		ct_srv_src	dst_host_srv_diff_host_rate
18		ct_srv_dst	dst_host_srv_serror_rate
19		ct_dst_src_ltm	dst_host_rerror_rate
20		service	dst_host_srv_rerror_rate

**Table 3 sensors-23-04141-t003:** Data split.

Dataset	Data Split	Normal Data	Attack Data
WSN-DS	Training set	302,921	0
	Evaluating set	29,116	29,116
UNSW-NB15	Training set	1,862,200	0
	Evaluating set	75,691	75,691
KDD99	Training set	550,652	0
	Evaluating set	262,152	262,152

**Table 4 sensors-23-04141-t004:** Confusion matrix.

Attack_label			Predict_label
		Attack	Normal
True_label	Attack	TP	FN
	Normal	FP	TN

**Table 5 sensors-23-04141-t005:** Loss and model size information.

Dataset	Loss	Model_Size
WSN-DS	0.017	338 KB
UNSW-NB15	0.012	1.1 MB
KDD99	0.008	423 KB

**Table 6 sensors-23-04141-t006:** Confusion matrix of WSN-DS evaluating set.

Attack_label			Predict_label
		Attack	Normal
True_label	Attack	28,693	423
	Normal	793	28,323

**Table 7 sensors-23-04141-t007:** Confusion matrix of UNSW-NB15 evaluating set.

Attack_label			Predict_label
		Attack	Normal
True_label	Attack	74,189	1502
	Normal	135	75,556

**Table 8 sensors-23-04141-t008:** Confusion matrix of KDD99 evaluating Set.

Attack_label			Predict_label
		Attack	Normal
True_label	Attack	257,505	4647
	Normal	4656	257,496

**Table 9 sensors-23-04141-t009:** Metrics.

Evaluating Set	Accuracy	Precision	Recall	F1_Score
WSN-DS	0.9791	0.9792	0.9854	0.9792
UNSW-NB15	0.9892	0.9893	0.9802	0.9891
KDD99	0.9823	0.9823	0.9823	0.9823

**Table 10 sensors-23-04141-t010:** Comparison of several NIDS.

Dataset	Approach	Accuracy	Precision	Recall	F1_Score
WSN-DS	SVM [44]	0.96	-	-	-
	CNN [45]	0.97	-	-	-
	Software-defined [46]	0.97	-	-	-
	Our approach	0.9791	0.9792	0.9854	0.9792
**Dataset**	**Approach**	**Accuracy**	**Precision**	**Recall**	**F1_score**
UNSW-NB15	AC-GAN [6]	0.96	0.96	0.98	0.97
	CAE and OC [14]	0.94	-	-	0.95
	Emsemble [33]	0.9523	0.9658	0.9594	0.9623
	Our approach	0.9892	0.9893	0.9802	0.9891
**Dataset**	**Approach**	**Accuracy**	**Precision**	**Recall**	**F1_score**
KDD99	CAE and OC [14]	0.9158	-	-	0.9287
	AE and SVM [25]	0.9472	-	-	-
	Stacked AE [21]	0.9817	0.9918	0.9522	0.9715
	Our approach	0.9823	0.9823	0.9823	0.9823

**Table 11 sensors-23-04141-t011:** Attack data split.

Dataset	Anomaly Data	Grayhole	Blackhole	TDMA	Flooding (0-Day Attack)
WSN-DS	Training set	10,063	5393	5312	0
	Evaluating set	2539	1374	1278	0
	Novelty set (0-Day Attack Set)	0	0	0	3157
	Total	12,602	6767	6590	3157
**Dataset**	**Anomaly Data**	**Dos**	**Probe**	**Access**	**Privilege (0-Day Attack)**
KDD99	Training set	197,767	11,106	808	0
	Evaluating set	49,493	2736	191	0
	Novelty set (0-Day Attack Set)	0	0	0	51
	Total	247,260	13,842	999	51
**Dataset**	**Anomaly Label**	**Training set**	**Evaluating set**	**Novelty set (0-Day Attack Set)**	**Total**
UNSW-NB15	Exploits	20,415	4979	0	25,394
	Fuzzers	14,881	3809	0	18,690
	Generic	13,695	3492	0	17,187
	Reconnaissance	6752	3809	0	8410
	DoS	2894	723	0	3617
	Shellcode	1164	282	0	1446
	Analysis	352	89	0	441
	Backdoor	273	74	0	347
	Worms (0-day attack)	0	0	159	159

**Table 12 sensors-23-04141-t012:** Ensemble model on training set and evaluating set.

Dataset	Method	Dataset_Type	Accuracy	Precision	Recall	F1_Score
WSN-DS	lightGBM	Training set	0.9971	0.9971	0.9971	0.9971
		Evaluating set	0.9882	0.9884	0.9882	0.9883
	XGBoost	Training set	0.9986	0.9986	0.9986	0.9986
		Evaluating set	0.9909	0.991	0.9909	0.9907
	RandForest	Training set	0.9998	0.9998	0.9998	0.9998
		Evaluating set	0.9946	0.9946	0.9946	0.9946
	Soft-Voting	Training set	0.9995	0.9995	0.9995	0.9995
		Evaluating set	0.9934	0.9934	0.9934	0.9934
UNSW-NB15	lightGBM	Training set	0.9841	0.9843	0.9841	0.9836
		Evaluating set	0.9063	0.904	0.9063	0.9004
	XGBoost	Training set	0.9665	0.9672	0.9665	0.9657
		Evaluating set	0.9081	0.908	0.9081	0.9021
	RandForest	Training set	0.992	0.992	0.992	0.992
		Evaluating set	0.8888	0.8859	0.8888	0.8825
	Soft-Voting	Training set	0.9883	0.9884	0.9883	0.9881
		Evaluating set	0.9074	0.9065	0.9073	0.9011
KDD99	lightGBM	Training set	0.9999	0.9999	0.9999	0.9999
		Evaluating set	0.9999	0.9999	0.9999	0.9999
	XGBoost	Training set	0.9999	0.9999	0.9999	0.9999
		Evaluating set	0.9999	0.9999	0.9999	0.9999
	RandForest	Training set	0.9999	0.9999	0.9999	0.9999
		Evaluating set	0.9999	0.9999	0.9999	0.9999
	Soft-Voting	Training set	0.9999	0.9999	0.9999	0.9999
		Evaluating set	0.9999	0.9999	0.9999	0.9999

**Table 13 sensors-23-04141-t013:** Ensemble model on novelty set (0-Day Attack Set).

Dataset	0-Day Type	Total	Method	Dos	Probe	Access
WSN-DS	Flooding	3157	lightGBM	1513	31	1613
			XGBoost	2739	13	405
			RandomForest	2592	29	176
			Soft-Voting	2591	18	548
KDD99	Privilege	51	lightGBM	0	0	51
			XGBoost	0	0	51
			RandomForest	0	0	51
			Soft-Voting	0	0	51

**Table 14 sensors-23-04141-t014:** Ensemble model on novelty set (UNSW-NB15).

Novelty Type	Recognition Label	LightGBM	XGBoost	RandomForest	Soft-Voting
WORMS	Exploits	136	134	117	136
	Fuzzers	12	12	12	12
	Generic	11	12	28	11
	Reconnaissance	0	0	0	0
	DoS	0	1	0	0
	Shellcode	0	0	2	0
	Analysis	0	0	0	0
	Backdoor	0	0	0	0
	Total	159	159	159	159

## Data Availability

Data is unavailable due to privacy.

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
