# Peer review of "A Lightweight Intelligent Network Intrusion Detection System Using One-Class Autoencoder and Ensemble Learning for IoT"

_sensors, 2023, doi:10.3390/s23084141_

Round 1

Reviewer 1 Report

The paper proposes a One-Class recognition method based on a Bidirectional GRU Autoencoder, which enables accurate recognition of normal and abnormal data, and ensures that the model trained with normal data has high prediction accuracy in abnormal processes. moreover in this paper, a multi-classification recognition method based on ensemble learning is also proposed, which uses soft voting technology to comprehensively judge the results of various base classifiers, and identifies unknown attacksNovelty data as the most similar type to known attacks, so as to make exception classification more accurate. The authors show that their approach  Experiments are conducted on WSN-DS, UNSW-NB15 and KDD CUP99 dataset, and the recognition rates of the proposed models in the three dataset are improved, reaching 97.91%, 98.92% and 98.23% respectively, which verifies the feasibility, efficiency and portability of the algorithm proposed in the paper. The literature study in the paper shows clearly what is missing in Intrusion Detection System for IoT. The paper is well written and well structured and the topic is highly relevant.
Nevertheless, I have three main concerns:

1. The research domain has not been clearly defined in the article. It is better to classify the literature review before the literature review section and define the research domain using a tree diagram.

2. At the end of the literature review section, it is better to compare different methods using a table and identify the strengths and weaknesses of each of the methods examined.

3. The quality of the article in terms of grammar is acceptable, however, there are some issues in certain parts that need to be corrected. For example, in line 113, "its" should be corrected to "it's". Additionally, in lines 213 and 216, "udp", "icmp", "tcp", "ftp", "http", "ipv6", and "dns" should be corrected to "UDP", "ICMP", "TCP", "FTP", "HTTP", "IPv6", and "DNS". 

In line 126 and 129, Say explicitly Equation (x) instead of following formula to make it easier to understand what you are referring to.

In line 240, it is better to use the phrase "in eq. 10" instead of "in this formula".

The titles of figures 12, 13, 14, and 15 are placed under their respective figures, which has made the titles difficult to read.

Reviewer 2 Report

Comments to Author

The authors proposed One-Class recognition method based on Bidirectional GRU Autoencoder for novelty detection (OC-Bi-GRU-AE and EL) model in network intrusion detection systems. The paper in current form is not qualify for publishing in the sensors journal because of the following:

1) The English writing is not clear, and it needs extensive enhancement, for example the content of Abstract is not understandable, many sentences are long and complex writing with wrong English structure.

2) The introduction is very poor and the information in the introduction is not referenced. Furthermore, the content missing the overview about the IoT attacks, NIDS and NIPS, challenges of NIDS in IoT, etc.    

3) The contribution is very small, and it should be extended to include the comparison with the recent works in NIDS.

4) The algorithm is very sample, and it should include more details about the proposed system.

5) The result section was written as a technical report not as an academic soundness paper. The authors should compare with recent methods that used the One-Class recognition method in NIDS not just compare with available dataset.

Round 2

Reviewer 2 Report

Comments to Author

The authors addressed my comments in first round very well, however, there are still some small comments:

1) In the second sentence of the abstract, authors should justify why the " It is hard for traditional intrusion detection systems with Binary or Multi-Classification as 2 targets to resist unknown attacks (such as zero-day attacks). "

2) Section 4.3 metrics should be integrated with section 4.2 which should be summarized.

3) English Language needs further enhancement.
